# Is Causality a Necessary Tool for Understanding Our Universe, or Is It a Part of the Problem?

**DOI:** 10.3390/e23070886

**Published:** 2021-07-13

**Authors:** Martin Tamm

**Affiliations:** Department of Mathematics, University of Stockholm, 106 91 Stockholm, Sweden; matamm@math.su.se

**Keywords:** entropy, causality, EPR paradox, accelerating expansion, time’s arrow

## Abstract

In this paper, the concept of causality in physics is discussed. Causality is a necessary tool for the understanding of almost all physical phenomena. However, taking it as a fundamental principle may lead us to wrong conclusions, particularly in cosmology. Here, three very well-known problems—the Einstein–Podolsky–Rosen paradox, the accelerating expansion and the asymmetry of time—are discussed from this perspective. In particular, the implications of causality are compared to those of an alternative approach, where we instead take the probability space of all possible developments as the starting point.

## 1. Introduction

Should causality be viewed as a fundamental property of nature? What causality really stands for is a question that goes back to ancient times, at least to Aristotle. However, even after 2500 years of discussions, no conclusive answer has been given, so indeed it appears to be a truly difficult one. What *is* clear, however, is that causality is absolutely indispensable for every kind of human activity, including the development of science, which gives an obvious argument in favor of attributing a fundamental status to it. Since modern physics started to develop in the beginning of the twentieth century, causality has been an important issue in both quantum mechanics and in the theory of relativity. In fact, to many scientists, causality is an even more fundamental concept than, for example, time (see [1]). On the other hand, this argument can, of course, equally be turned around to say that the very fact that we are evolutionarily so well trained to interpret everything in term of cause and effect may be the reason as to why it is so hard for us to see the alternatives.

So what could be wrong about using the concept of causality, which, after all, serves us so well in everyday life? It is the purpose of this paper to present some examples of what could actually go wrong. All of these examples will appear in a highly simplified form to make them as accessible as possible, and in no case is the discussion here claiming to assert any kind of final answer to the specific problems.

IThe first example is concerned with the possibility that the concept of causality may itself create problems, which, from another point of view, would be pseudo-problems or no problems at all. As an example, the long-standing discussion of the Einstein–Podolsky–Rosen paradox (EPR) is considered; see Section 3.IIThe second example discusses the possibility that we may miss fundamental components in our theories if we insist that the future should exclusively be considered as a consequence of the past. In this case, the example is the accelerating expansion of our universe; see Section 4.IIIThe third example is about the direction of time. When we talk about cause and effect, we always suppose that the cause should precede the effect in time, but why? Complete time symmetry is obviously incompatible with causality since the roles of cause and effect would then also be symmetric. So could it be that the idea of causality actually prevents us from seeing the underlying symmetry of time? This is the theme in Section 5.

It is obvious that any theory with the ambition to explain the world as we know it must somehow also explain causality. However, there is a very substantial difference between viewing causality as a fundamental concept and as an emergent concept. So could it be that we would simply do better in understanding the world on the deepest level without the idea of causality? The opinion of the author of this paper is that before accepting causality as a fundamental concept, we should at least investigate the alternatives. In the next section, such an alternative description of the world is discussed.

The purpose of this paper is to argue in favor of a different perspective on fundamental physics, rather than to present new facts in each of the particular examples discussed. For this reason, the number of technical details has been kept to a minimum. For the reader interested in a more detailed technical treatment, references are given in Section 4 and Section 5.

## 2. The World as a Probability Space

One of the most fruitful ideas of all time in mathematical physics is the idea of an initial value problem. This idea was well established by Newton. If we know the positions and the velocities of all bodies in our problem at a certain time $t_{0}$ as well as all the forces acting on the bodies, then Newton’s second law can be used to set up differential equations to determine the positions of the bodies at any time in the future.

It goes without saying that this idea has a very strong concept of causality built into it: the initial conditions are exactly what we need to predict the future, and from this, it is only a small mental step to the conviction that the future is actually *caused* by the present. In any case, the idea has turned out to be tremendously successful. It is hard to imagine anything, such as the technological development since Newton’s time, without it.

Therefore, it must be considered very natural that, when quantum physics was discovered, the laws of motion of the new theory (e.g., Schrödinger’s equation) were formulated in a way that is essentially just as deterministic as Newton’s equations. However with a twist: even if the initial state of the wave function determines its future development perfectly, it is impossible to fully know the initial conditions and, similarly, it is impossible to know the final state. This is so for very fundamental reasons. Thus, we are left with a theory that “in principle” is completely deterministic but in practice is stochastic.

In the history of physics, there is also a parallel track: we can consider the set of all possible histories and then try to find out what characterizes the one that we actually observe. This is the perspective of Maupertius [2], with his principle of least action, but it certainly goes further back, at least to Fermat. However, in the early history of this approach, there was a wide gap between the real history and all the other (non-existing) histories. A major step toward a different point of view was taken with the idea of “democracy of all histories” (mainly due to Feynman (see [3])). In this approach, all possible histories are considered to be equally real, but the dominating ones result from superposition and interference between all candidates. The main problem of physics then becomes to compare the probabilities of all these developments and to decide which developments are the most likely ones. As it miraculously turns out, in the classical limit, this formulation becomes essentially equivalent to the classical initial value problem approach.

This idea has been further developed in many different ways. One radical step was taken when the idea of the “multiverse” was introduced. The multiverse concept was quickly welcomed by many since this idea appeared to resolve one of the most disturbing problems with the Copenhagen interpretation: the very special role of the human observer. In recent years, however, some of its popularity seems to have faded away. This may partly be because the idea has been over-used: the multiverse has sometimes become an arena where everything is possible, which is not very useful for making firm predictions. In addition, the rapid development of “QBism”, Bayesian quantum mechanics (see [4]) may have made the role of the human observer appear more natural.

In this paper, we are concerned with what may be called “pre-multiverses”. This should not be thought of as a purely quantum mechanical concept. Rather, the name should be considered a manifestation of the idea that the basic objects of study in physics are probability spaces, consisting of all possible developments. This is an idea on a more basic level, which, in principal, makes sense both in a classical and a quantum mechanical setting. There is, however, one important property to be noted: the dynamics here will *not* automatically be deterministic in the Newtonian sense. Thus, a specific state at a given time may, in general, develop into several different states at another time. This does not, of course, mean that the future should be wholly unpredictable. However, this view does try to incorporate the fact that some physical processes, e.g., radioactive decay, are, in principle, impossible to predict and that the outcome of such processes may lead to completely different futures within a relatively short time.

So why should it matter whether we adopt a causal perspective, where the future is regarded as a consequence of the past, or a probability space perspective, where causality plays no fundamental role but rather the developments that occur are determined by general timeless principles? After all, in most situations in common physics, these perspectives are equivalent.

To answer this question, it is reasonable to look at situations where this equivalence may not apply. In particular, this concerns the global properties of the universe (see Section 4 and Section 5). As it turns out, in such situations, the differences can be very substantial. Before we proceed, let us, as an example, consider a very well-known issue in quantum mechanics.

## 3. The EPR Paradox

So much has been said and written about the EPR paradox that it is difficult to add something new to the discussion. Here, it will mainly serve as an introduction to the probability space approach.

In modern language, the Einstein–Podolsky–Rosen paradox, in its most elementary form, is concerned with two entangled particles with opposite spin. We do not know the spins of the individual particles, only that they were both created at time $T_{1}$ with total spin equal to zero; once we measure the spin of one of the particles at time $T_{2}$, we can immediately know the spin of the other particle, even if they are separated by a very large distance. So it appears as though the influence of the measurement on the first particle instantaneously reaches the second particle, which contradicts the principles of the theory of relativity since such an influence should not be able to travel faster than light.

Is this a paradox? I will not here go into all the different arguments that have been presented. Let us just note that, according to the philosophy of the present paper, we should not consider it to be one: the paradox arises only when we try to interpret the experiment in terms of causality. In terms of the probability space approach, what happens instead is that, in the probability space Ξ of all possible developments of the two particles, starting with the creation at $T_{1}$ and ending with the measurement at time $T_{2}$, the probabilities for the events are as follows:(1)$$E_{+-}:\mathrm{particle}1↑,\;\;\mathrm{particle}2↓,$$
(2)$$E_{-+}:\mathrm{particle}1↓,\;\;\mathrm{particle}2↑,$$
are both ≈50%, whereas the probabilities for the events
(3)$$E_{++}:\mathrm{particle}1↑,\;\;\mathrm{particle}2↑,$$
(4)$$E_{--}:\mathrm{particle}1↓,\;\;\mathrm{particle}2↓,$$
are both ≈0%.

When we measure particle 1, what happens is not that this “effects” the state of the second particle. What happens instead is that the measurement is a fork in the road toward the future, and the outcome of the measurement tells us which road we and the two particles have taken.

From the probability space point of view, there is nothing particularly quantum mechanical (nor classical for that part) about this point of view: it is just a restatement of the conservation of spin. However, since much of the discussion about the EPR paradox has been exactly about the difference between classical physics and quantum physics, this may deserve a comment.

Again, from the probability space view point, the more or less classical explanation that stems from the original paper [5] is that the bifurcation actually takes place at time $T_{1}$, and it is only our knowledge of the result that appears at time $T_{2}$. However, this point of view does not appear to be compatible with quantum mechanics (see [6]): what actually makes the world quantum mechanical is that we do not, in fact, live in just one history but in several. Nearby histories do interact with each other, e.g., when a particle simultaneously passes through two slits and afterwards interacts with itself. According to this point of view, the events $E_{+-}$ and $E_{-+}$ may coexist in our world, and the fork in the road does, in fact, appear exactly when the measurement is made at time $T_{2}$. How and when different histories interact with each other may in itself be considered one of the mysteries of quantum mechanics that we still do not completely understand. However, this is a very different kind of question that is not very closely related to the question of causality.

## 4. The Global Geometry of Our Universe

As has already been remarked, for almost all practical purposes, the initial value approach and the probability space approach lead to equivalent results when applied to ordinary physical problems. This fact, of course, influenced the early development of modern cosmology. It can be said that the field equations of general relativity may not fit quite as well into the paradigm of initial value problems as do Newton’s and Schrödinger’s equations. Nevertheless, this is exactly the way that they have almost always been treated since they were discovered. In fact, the historically most important question of modern cosmology has been whether or not there is enough matter in our universe to stop the expansion and make it contract into a Big Crunch. In other words, we want to use the initial conditions that we observe right now to compute the future of the universe, just as we would use Newton’s theory to decide whether or not a certain object that we launch will be able to escape the gravitational influence of the earth or fall back down to the surface.

In recent years, it has become mainstream to assume the extension of space-time to be infinite, due in large part to the discovery of the (accelerating) expansion [7,8]. The reason is exactly that the observations that we make when extrapolated into the future seem to lead to an ever-expanding universe, and the most common interpretation is that the field equations have to be modified, e.g., by introducing some kind of “dark energy”.

It has, however, turned out to be difficult to explain the nature of this dark energy, so it may be a good idea to remember that this is not the only way of posing the problem. From the probability space point of view, there is nothing obvious about the initial values determining everything on the cosmic scale. Instead, we should try to decide what the natural conditions of the probability space of all possible universes are. So, let us try to see what such an alternative approach could look like in a highly simplified form. Whether or not this way is better than the usual approach is not the issue here; the point is to argue that the idea of the present determining the future is not the only possible one.

The theory of relativity has changed our view of the world in many ways. In particular, the old view of space and time as an empty arena, where the real actors—massive bodies—perform their play, has had to stand back in favor of a view where space-time itself plays an active role, more or less on equal terms with matter. Starting from this point of view, it seems very natural to assume that space-time itself is built out of some kinds of microscopic entities, and that the number of such entities is a fixed (finite) number, which directly leads to the condition that the total four-volume of space-time is fixed. From such a starting point, one can, rather than viewing the development as an initial value problem, try to find the optimal shape of the universe, according to a principle of least action in some form.

It is by no means a trivial problem to formulate the total action of the universe, in particular since it must take into account not only ordinary mass–energy, but also the action of space-time itself. For the sake of the discussion here, however, let us consider only the geometry of space-time and the potential energy.

Consider, therefore, a closed model for the global geometry of the universe, where the radius a(t) of the spacial part starts from zero at the Big Bang and ends at zero at a future Big Crunch. Assuming homogeneity and isotropy, the form of a(t) gives a good description of the global geometry.

**Remark** **1.**
*One may, of course, also ask what the probability space perspective would lead to in the case of an unbounded model, where the total volume of space-time is infinite. This would, however, lead to considerable technical complications since, in this case, it would not be possible to treat the set of all possible geometries as a probability space. Rather, one would have to treat the global case as a kind of limit of the cases with finite space-time volume. This would be very much in the tradition of thermodynamics. Nevertheless, it would make the treatment here considerably more complicated. Hence, for the purposes of the present paper, it seems natural to concentrate on the (simpler) case of a closed model.*


For the geometric part, let us take the following action, depending only on the scalar curvature:(5)$$\int_{\Omega }R^{2}\;dV=\frac{\pi^{2}}{2}\int_{I}\frac{36{\left(1+a^{\prime }{\left(t\right)}^{2}+a\left(t\right)a^{\prime \prime }\left(t\right)\right)}^{2}}{a(t)}\;dt.$$

This is, in fact, a simplified form of the action analyzed in [9], where also more details are given.

For the potential energy, we take the classical Newtonian law of gravitation, which states that the potential energy between two objects is inversely proportional to the distance between them. In combination with homogeneity, this leads to the following action term:(6)$$-\int_{I}\frac{1}{a(t)}\;dt.$$Combining these two terms, we thus arrive at the following action:(7)$$L=\frac{\pi^{2}}{2}\int_{I}\frac{36{\left(1+a^{\prime }{\left(t\right)}^{2}+a\left(t\right)a^{\prime \prime }\left(t\right)\right)}^{2}}{a(t)}\;dt-\beta \int_{I}\frac{1}{a(t)}\;dt,$$
where β>0 is some constant, which may be hard to determine. Assuming that the total volume is fixed, the classical calculus of variation can now be applied by making use of the Euler–Lagrange equation to analyze the shape of the universe in terms of a(t). This is a huge task, and I will not go into details here. In Figure 1 is plotted a more or less typical solution.

The reader may note that, in this picture, there is a phase of accelerating expansion (and a corresponding period of accelerated contraction). In contrast to other current theories, this is not the result of introducing some extra field or dark energy. Instead, it is the finiteness of the 4-volume itself that makes these phases necessary.

Rather than going deeper into the technical details behind Figure 1, I will try to give an intuitive explanation as to why the universe should take such a form. The main point to observe is that if we start with an empty, massless closed universe, the natural shape of such a universe in relation to the above action is not flat space-time, but a 4-sphere. In fact, a direct computation shows that the scalar curvature of such a sphere is zero.

So what happens to such a 4-sphere when matter is introduced? It is intuitively clear that this will give rise to an attractive force which is strongest when close to the Big Bang and the Big Crunch, and this force will deform the universe by contracting the spacial part near the endpoints. However, since the volume is fixed, this contraction near the ends must somehow be compensated by an expansion in the middle (where the contracting force is weaker) (see Figure 2), thus resulting in behavior similar to that in Figure 1.

Needless to say, this argument should not in itself be considered a rigorous scientific argument. The point is that it gives intuition to the results obtained by the Euler–Lagrange method above. For more details, see [9,10].

Summing up, the assumption about fixed volume could give an alternative explanation for the accelerating expansion without any need to introduce extra components, such as dark energy, into the field equations. However, this explanation obviously does not make sense in a paradigm where the starting point is to view the development of our universe as an initial value problem. In fact, in the variational problem, which determines the shape of the universe in this section, the past and the future play equally important roles.

## 5. The Asymmetry of Time

The asymmetry of time is one of the great mysteries in science. On the micro-level, the laws of nature appear to be essentially time symmetric: if a process is possible in one direction of time, then the reversed process is also possible. On the macro-level, however, this is no longer true. In fact, in our world as we know it, reversible processes are very rare. So where does this asymmetry on the macro-level come from?

Few questions have generated such a large number of completely different answers (see [11,12]) but we are still very far from reaching an agreement. It is not very easy to give an exact definition of what macroscopic time asymmetry means. We all know that yesterday is very different from tomorrow, for example, because we can remember yesterday but not tomorrow, but how do we put this into physical and measurable terms? Here, I will avoid all such problems by simply concentrating on the most obvious thermodynamic aspect of time asymmetry as manifest in the second law of thermodynamics.

**Time’s** **Arrow.**
*The direction toward the future is the direction in which the total entropy of our universe increases, and the direction toward the past is the direction in which the entropy decreases.*


So how can one attempt to explain time asymmetry?

The most common explanation of the second law consists of the idea that for some reason, our universe started from a state with extremely low entropy. This goes back to Boltzmann’s ingenious idea that the second law of thermodynamics is a manifestation of the fact that our universe developed from less probable states (i.e., states with low entropy) to more probable states (i.e., states with higher entropy) simply because there are so many more of the latter kind (see [13]). Certainly, in a way, this argument is very convincing, but from the cosmological point of view it becomes more problematic, as it just deduces one kind of time asymmetry by assuming another kind (different boundary conditions in the past and in the future). In fact, the mere statement that the universe starts from some state without assuming that it ends at a similar state builds a kind of time asymmetry into the model (see Price [14]).Another way of trying to explain time asymmetry is to assume that the dynamical laws, although they appear to be very time symmetric, still have something asymmetric built into them. This is the approach of Penrose, who tried to connect the asymmetry with the behavior of the Weyl tensor, which is the non-observable part of the Riemann tensor (see [15]). It is also the approach of Sakharov, who attempted to use the apparently time-asymmetric behavior of the K meson to explain time asymmetry (see [16,17]).A third way is to assume that our universe *is* time symmetric by connecting the growth of entropy with the growth of space itself. In such a universe, entropy would start to decrease when the universe starts to contract, and the world would end up in a Big Crunch with levels of entropy as low as the Big Bang (see Gold [18]).

All of these ideas have their own relation to the concept of causality, and there are also many other attempts. I will not go into the details here but note that, so far, none of these ideas has been able to convince the scientific community.

However, if we are prepared to give up the idea of causality completely on the global scale (in favor of the probability space approach), a fourth possibility appears. The point of view to be advocated here is that the world may, in a certain sense, be time symmetric, even on the macro-level; it is just that our very human perspective prevents us from seeing it. It could very well be that the space of all possible universes is time symmetric and governed by completely time symmetric dynamical laws. Even if this were the case, in almost every individual universe (and, hence, to each observer) time would be highly asymmetric.

It goes without saying that it is impossible to model the set of all possible universes with any detail. What can be done, however, is to study very small models for this probability space. Here, I will briefly sketch one such model. We start from the very simplified but symmetric boundary conditions at the Big Bang and the Big Crunch: we might say that both of them correspond to unique states with zero volume and zero entropy. In between, we consider time to be discrete, and for each such discrete moment of time, there is a finite number of possible states. The dynamics of the model is then given by specifying which passages are possible from a given state at a certain time *t* to other states at the next or previous moment of time.

What we obtain in this way is an enormous graph, where the nodes are the states and the edges are the possible passages between states at adjacent moments of time.

**Definition** **1.***A* universe Ξ *is a path in this graph, i.e., a chain of states from the Big Bang to the Big Crunch, one state $\Xi_{t}$ for each moment of time t, with the property that the transition between adjacent states is always possible.*

**Definition** **2.***The* pre-multiverse *M is the set of all possible universes* Ξ *in the sense of Definition 1.*

To make such a large model possible to handle, we must now forget about almost everything that we know about physics, and just keep one concept: entropy *S*. The basic idea about the dynamics essentially goes back to Boltzmann but now appears in symmetrized form:

**Assumption** **1.***(The symmetric Boltzmann principle.) For any state* Ξ *with entropy S at a certain time t, there are many edges connecting* Ξ *to different states with entropy larger than S at the next moment of time; similarly, there are many edges connecting* Ξ *to states with entropy higher than S at the previous moment of time. However, in both directions of time, the chances of finding a continuation leading to a state with entropy lower than S are very poor.*

These dynamics and boundary conditions can be used to turn the set of all possible universes into a probability space. So what does the most probable kind of universe look like? It turns out that under quite general conditions of the dynamics, as those in Assumption 1 above, monotonic entropy (either growing or decreasing) will vastly dominate over universes with low entropy at both ends, thus, in a sense, establishing at least a weak form of the second law over the whole lifespan from the Big Bang to the Big Crunch; see Figure 3. An intuitive motivation for this could go as follows: with the dynamics in Assumption 1, a switch from increasing to decreasing, or from decreasing to increasing, behavior of entropy is an extremely unlikely event. The only place where it could have a reasonable chance to occur would be very shortly after the Big Bang or immediately before the Big Crunch, where the total volume is so small that quantum effects can be expected to be global and, hence, in a sense, transitions between any states are possible.

For stronger versions of the second law, extra conditions on the dynamics may be needed but there are many open questions here.

Much more can be said about such models. For a more comprehensive discussion, the reader is referred to [19,20,21,22]. To stress, these models are exaggeratedly small and over-simplified, compared to the real world. Thus, from a cosmological point of view, much more research is needed. Nevertheless, what is said here may be enough to illustrate the point of the probability space perspective as compared to the usual causal point of view.

## 6. Conclusions

When summarizing the content of this paper, a good start may be to summarize what is *not* said.

Is causality a worthless concept in cosmology?No, definitely not. However, we must remain open to the possibility that it is not a fundamental principle on the global scale.Is the kind of global condition used in Section 4 to be considered a kind of backward causation? (See [23].)No. But from the point of view of the author, it may in some (rare but important) situations simply be better to leave out causality, both forward and backward causation alike.Is this a quantum theory or a classical theory?Both—or neither. There is no doubt that quantum theory is a better description of reality that the classical one, but it may be that the idea of causality should be dealt with on an even more fundamental level.Is the theory in this paper a multiverse theory?Yes, and no. Certainly, there are similarities in the sense that the universe in which we find ourselves is not perceived as the only real one. However, recall that the idea of the “multiverse” can be said to have emerged naturally from the Everett interpretation (see [24]) and it is, therefore, a product of what can now be called traditional quantum mechanics. Very much at the heart of it is the idea of a deterministically developing wave-function. It is also very much in line with classical causality: the road towards the future may constantly bifurcate but in a perfectly deterministic way. The past, on the other hand, is perfectly unique, at least on the macroscopic scale. Thus, in this perspective, time asymmetry is very much built into the dynamical laws. In contrast, in the pre-multiverse, which I am suggesting here, the dynamics in itself is time symmetric but also non-deterministic, in the sense that a given state may lead to different states both in the future and in the past. That we still perceive our history as unique is instead, in this interpretation, a consequence of the fact that the number of possible universes starting from the Big Bang, although very large, is still extremely small compared to the number of possible states of a universe at the present time. This, in turn, means that the probability of two such different universes “meeting” later on at the same state is, for all practical purposes, zero. This is just another way of saying that when we look backwards in time, we can only see one history. For more details, see [25].

The list of examples in this paper is in no way complete. It simply reflects the author’s preferences, and perhaps his lack of knowledge of situations in which a causal and a non-causal approach may generate different perspectives on reality.

One may ask if it will ever be possible to test the two approaches against each other. In cosmology, the best that we can do is usually to start from general ideas and then deduce the consequences of these ideas that can be compared with reality. In this sense, predictions about, for example, the shape of our universe, as discussed in Section 4, are testable, even if a lot of work remains before these assumptions can be made realistic enough.

It should also be remembered that the non-causal approach, in a sense, can offer a broader perspective than the causal one: *if*, for example, the condition of constant volume, as discussed in Section 4, turns out to have a real impact on the shape of our universe, then it would be hard to see how we could have discovered it without giving up causality. On the other hand, there is no evident reason as to why allowing for such a generalized point of view would stop us from investigating other possibilities, including the causal ones.

## Figures and Tables

**Figure 1 entropy-23-00886-f001:**
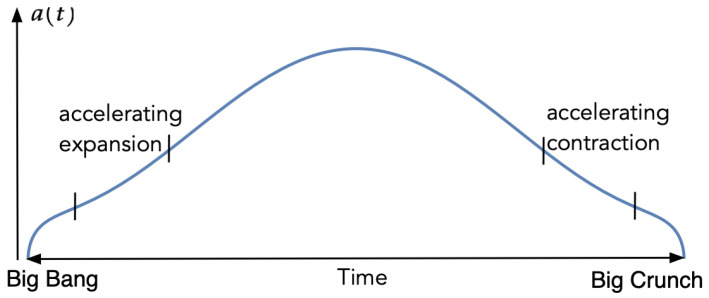
A solution of the Euler–Lagrange equation.

**Figure 2 entropy-23-00886-f002:**
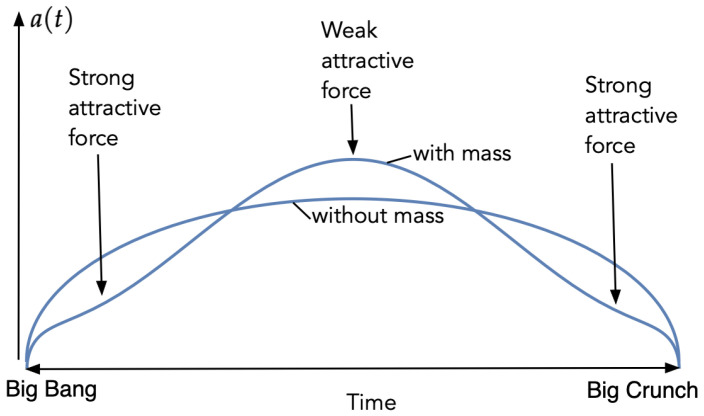
The effect of mass on the radius a(t), compared to the case without mass.

**Figure 3 entropy-23-00886-f003:**
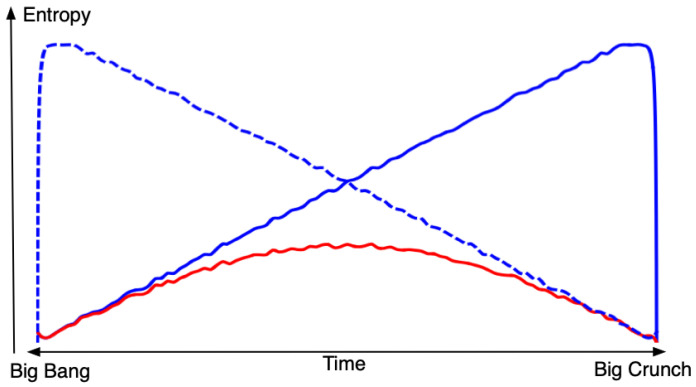
A schematic picture of the different possibilities for the entropy of a universe. The blue lines represent the highly probable types, whereas the red one represents a very unlikely kind of universe (of Gold’s type).

## Data Availability

Not Applicable.

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
