# Peer review of "Is Causality a Necessary Tool for Understanding Our Universe, or Is It a Part of the Problem?"

_entropy, 2021, doi:10.3390/e23070886_

Round 1

Reviewer 1 Report

The author considers causality in spacetimes, which is one of the important notions both in theoretical and experimental physics. The author introduces the history of causality with some unsolved problems in the section 1, and considers an idea of the probability space, which is related to the Heisenberg and the Dirac pictures in quantum mechanics, in the section 2. The author applies such an idea to the Einstein-Podolsky-Rosen paradox in the section 3, as an example in a microscopic system. Motivated by the concept of entropy and the second law of thermodynamics, the author assumes an existence of global time symmetry in the Universe and applies this idea to a time evolution of the Universe, as an example in a macroscopic system in the sections 4 and 5. Then the author obtains an interesting cosmological model that describes an accelerating expansion without a dark energy component. I find this work interesting but I recommend the author to consider a following point:
i) It seems that a cosmological model in the section 4 is obtained by assuming a closed universe. Then, can one obtain such a model in a flat universe and an open one? Is such an extension of the present solution difficult? The author may mention these things in the section 4.

Author Response

Answer to Ref 1.

Thank you for your valuable criticism. I have tried to comment on the cases with flat and open universes in Section 4. Basically, I do think that the probability space perspective should be applicable there too. But yes, it will definitely be  more  difficult, essentially since if  space-time has infinite  volume, then one can not really consider the set of all possible metrics as a probability space. Rather, one would have to construct a kind of limit of the probability spaces corresponding to subsets with finite volume. This would be very much in the spirit of thermodynamics as in the book of David Ruelle (Thermodynamic Formalism). But in the same time, it would be very technical and perhaps rather messy, at least  in the beginning before the right formulation has been found.

Martin Tamm

Reviewer 2 Report

The paper is devoted to the discussion about some fundamental questions in theoretical physics, particularly the need of causality and the confrontation of quantum mechanics principles vs classic ones. The whole paper is very speculative, written in more philosophical way than physical, which makes it very entertaining and might be of some interest to be read for non-experts but lacks of the necessary depth and details for a scientific-technical journal. Hence, I can not recommend the paper to be published in Entropy but it might suit better in other journals more focused on philosophical speculative debates.

Author Response

Answer to Ref 2.

Thank you for your valuable criticism. I do see your point. However, from my point of view, the situation is precisely the reversed one. In fact, if we discard the EPR part (which is just an introductory example), the other two problems are exactly what I have spent the last ten or fifteen years on trying to make as rigorous as I can, and I have also published approximately 10 papers with related material. May I ask you to take a brief look at e.g. the following papers?

https://doi.org/10.3390/sym8030011

https://doi.org/10.4236/jmp.2015.63029

https://doi.org/10.3390/universe7030074

The purpose of this paper is to try to extract something common and more general out of all the others, and this is exactly  why I have tried to concentrate on the  underlying common ideas and leave out as much as I can of the technical details. If I may add a personal remark, I can say that I have been rather frustrated during these years by the fact that the only kind of respons I usually get is about technicalities, but no one wants  to take any interest in the bigger perspective.

It is of course not my decision what kind of journal Entropy wants to be. If this paper is  considered to be too general or too philosophical, I will certainly accept that. If, on the other hand, the Editors do want to be open to this kind of papers, I would very much appreciate your suggestions about how I could make it clearer to the readers that the technical treatment is to be found elsewhere. Clearly, this was not evident in the first version, which is then of course my fault. I have added  a comment about this  in the introduction, but maybe something more could be done? Alternatively, if you consider this paper to be better suited for some other journal, which one would you recommend?

Let me also finally stress that, in spite of all my efforts, I do not consider myself to have any kind of final solutions to these problems. In fact, both problems are obviously so hard that only cooperation within the scientific community could have a chance to solve them. For example,  if my ideas about time asymmetry are anything close  to being correct, then  the only way forward which I can see would be massive computer simulations, which it is far beyond my own capacity to carry out by myself. So how am I supposed to go on if respectable journals only want to publish technical  aspects but not more general aspects? The target group which I’m trying to reach is not philosophers, nor is it physicists who only take interest in technical questions,  but the group of people who share both interests.

Martin Tamm

Reviewer 3 Report

Although the paper introduces some nice ideas, it is unable to provide any rigorous justification. The ideas are speculative and the author tries to reason all the grand challenges in theoretical physics through just one paper. I would encourage the author to seriously reconsider the current approach, and take one of the problems listed in the current paper and instead tackle it rigorously first before moving onto the other. Without rigor this paper does not add anything new to the existing literature on causality. Therefore, I cannot recommend to publish this article in its current form. 

Author Response

Answer to Ref 3.

Thank you for your valuable criticism. I do see your point. However, from my point of view, the situation is precisely the reversed one. In fact, if we discard the EPR part (which is just an introductory example), the other two problems are exactly what I have spent the last ten or fifteen years on trying to make as rigorous as I can, and I have also published approximately 10 papers with related material. May I ask you take a brief look at e.g. the following papers?

https://doi.org/10.3390/sym8030011

https://doi.org/10.4236/jmp.2015.63029

https://doi.org/10.3390/universe7030074

I do not think that it is fair to say that I try to solve all the big problems in just one paper. Rather, the purpose of this one is to try to extract something common and more general out of all the others, and this is exactly  why I have tried to concentrate on the  underlying common ideas and leave out as much as I can of the technical details. If I may add a personal remark, I can say that I have been rather frustrated during these years by the fact that the only kind of respons I usually get is about technicalities, but no one wants  to take any interest in the bigger perspective.

It is of course not my decision what kind of journal Entropy wants to be. If this paper is  considered to be too general or too philosophical, I will certainly accept that. If, on the other hand, the Editors do want to be open to this kind of papers, I would very much appreciate your suggestions about how I could make it clearer to the readers that the technical treatment is to be found elsewhere. Clearly, this was not evident in the first version, which is then of course my fault. I have added  a comment about this  in the introduction, but maybe something more could be done?

Let me also finally stress that, in spite of all my efforts, I do not consider myself to have any kind of final solutions to these problems. In fact, both problems are obviously so hard that only cooperation within the scientific community could have a chance to solve them. For example,  if my ideas about time asymmetry are anything close  to being correct, then  the only way forward which I can see would be massive computer simulations, which it is far beyond my own capacity to carry out by myself. So how am I supposed to go on if respectable journals only want to publish technical  aspects but not more general aspects? The target group which I’m trying to reach is not philosophers, nor is it physicists who only take interest in technical questions,  but the group of people who share both interests.

Martin Tamm

Round 2

Reviewer 1 Report

I am satisfied with the corrections by the author.
I would like to recommend the publication of this work.

Reviewer 2 Report

I appreciate the reply and arguments given by the author for this second round. My criticisms were not focused on the correctness of the discussion of the paper but on its suitability for Entropy journal. As far as I understand, Entropy is a highly specialised journal focused on technical thermodynamical issues in a broad range of areas of expertise, while the paper is a compendium of speculative discussions of some fundamental physics ideas, which while interesting themselves, in my opinion do not suit for Entropy Journal. However, as stated by the author, the decision for accepting this type of papers in Entropy corresponds to the Editorial Board.   

Therefore, even though my recommendation is "Reconsider after major revision", this is because there is no option for "No response".

Reviewer 3 Report

I empathize with the author's response to my criticism. To some extent I even agree with the author however to move forward with this manuscript I feel the author still needs to add some rigor to be considered a serious paper. Without rigor the ideas presented in this paper unfortunately reduces to borderline philosophy or popular science at best. Atleast, if the author can think of a possible model and include them in text - theoretical or numerical as they state in their response letter - I would be curious and perhaps reconsider my decision.